# 3D genome mapping identifies subgroup-specific chromosome conformations and tumor-dependency genes in ependymoma

Ependymoma is a tumor of the brain or spinal cord. The two most common and aggressive molecular groups of ependymoma are the supratentorial *ZFTA*-fusion associated and the posterior fossa ependymoma group A. In both groups, tumors occur mainly in young children and frequently recur after treatment. Although molecular mechanisms underlying these diseases have recently been uncovered, they remain difficult to target and innovative therapeutic approaches are urgently needed. Here, we use genome-wide chromosome conformation capture (Hi-C), complemented with CTCF and H3K27ac ChIP-seq, as well as gene expression and DNA methylation analysis in primary and relapsed ependymoma tumors, to identify chromosomal conformations and regulatory mechanisms associated with aberrant gene expression. In particular, we observe the formation of new topologically associating domains ('neo-TADs') caused by structural variants, group-specific 3D chromatin loops, and the replacement of CTCF insulators by DNA hypermethylation. Through inhibition experiments, we validate that genes implicated by these 3D genome conformations are essential for the survival of patient-derived ependymoma models in a group-specific manner. Thus, this study extends our ability to reveal tumor-dependency genes by 3D genome conformations even in tumors that lack targetable genetic alterations.

Tumors of the central-nervous system (CNS) are the most common cancers in children aged 0-14 years and a leading cause of death during childhood[1–3]. Intracranial ependymomas are segregated on the basis of anatomic location (supratentorial versus infratentorial or posterior fossa) and further divided by DNA methylation and expression profiling into distinct molecular groups that reflect differences in the age of onset, gender predominance, response to therapy, and genetic aberrations that drive the disease[4–6]. The supratentorial ZFTA(C11orf95)-fusion associated group is characterized by recurrent complex chromothripsis events on chromosome 11 that lead to different types of ZFTA(C11orf95)-RELA fusion genes, which have been shown to drive tumorigenesis in this group of tumors[7,8]. In contrast, initial DNA sequencing studies showed an absence of recurrent mutations or gene fusions in posterior fossa ependymoma group A (PFA), suggesting that

these tumors might be epigenetically driven[5,7]. Indeed, global loss of histone H3 lysine 27 trimethylation (H3K27me3), a histone modification associated with the negative regulation of gene expression, was identified as a marker for PFA tumors[9]. Recent studies have revealed that EZH inhibitory protein EZHIP (previously known as *CXorf67*), which is aberrantly expressed in most PFA ependymomas (and mutated in some), causes downregulation of H3K27me3 by inhibiting EZH2 in the polycomb repressive complex 2 (PRC2)[10,11]. The few PFA ependymomas that do not overexpress *EZHIP* appeared to harbor K27M mutations in H3.1 or H3.3, which also inhibit EZH2. Furthermore, gain of chromosome arm 1q, present in ~25% of all PFA tumors, has been associated with a particularly poor survival of PFA patients, but the underlying driver mechanism remains unknown[10,12]. Since there are no small molecules available directly targeting the ZFTA fusions or EZHIP, and

✉e-mail: lukaschavez@health.ucsd.edu

since it is not yet known whether EZHIP alone drives tumorigenesis in PFA, a better understanding of the tumor driving mechanisms and how they can be targeted is urgently needed. New insights into the regulation of gene expression during normal and diseased human development have recently been gained by analyzing 3D chromatin architectures[13-15]. By combining Hi-C with complementary molecular profiling techniques on ependymoma tumors and cell lines, we identify lineage- and tumor- specific chromosomal DNA interactions in association with structural variants and tumor-dependency genes (Fig. 1a).

## Results

### The 3D genome organization of ependymoma tumors

We have performed Hi-C[16] followed by deep sequencing in 18 PFA and ZFTA ependymoma samples, comprising fifteen tumors (fresh

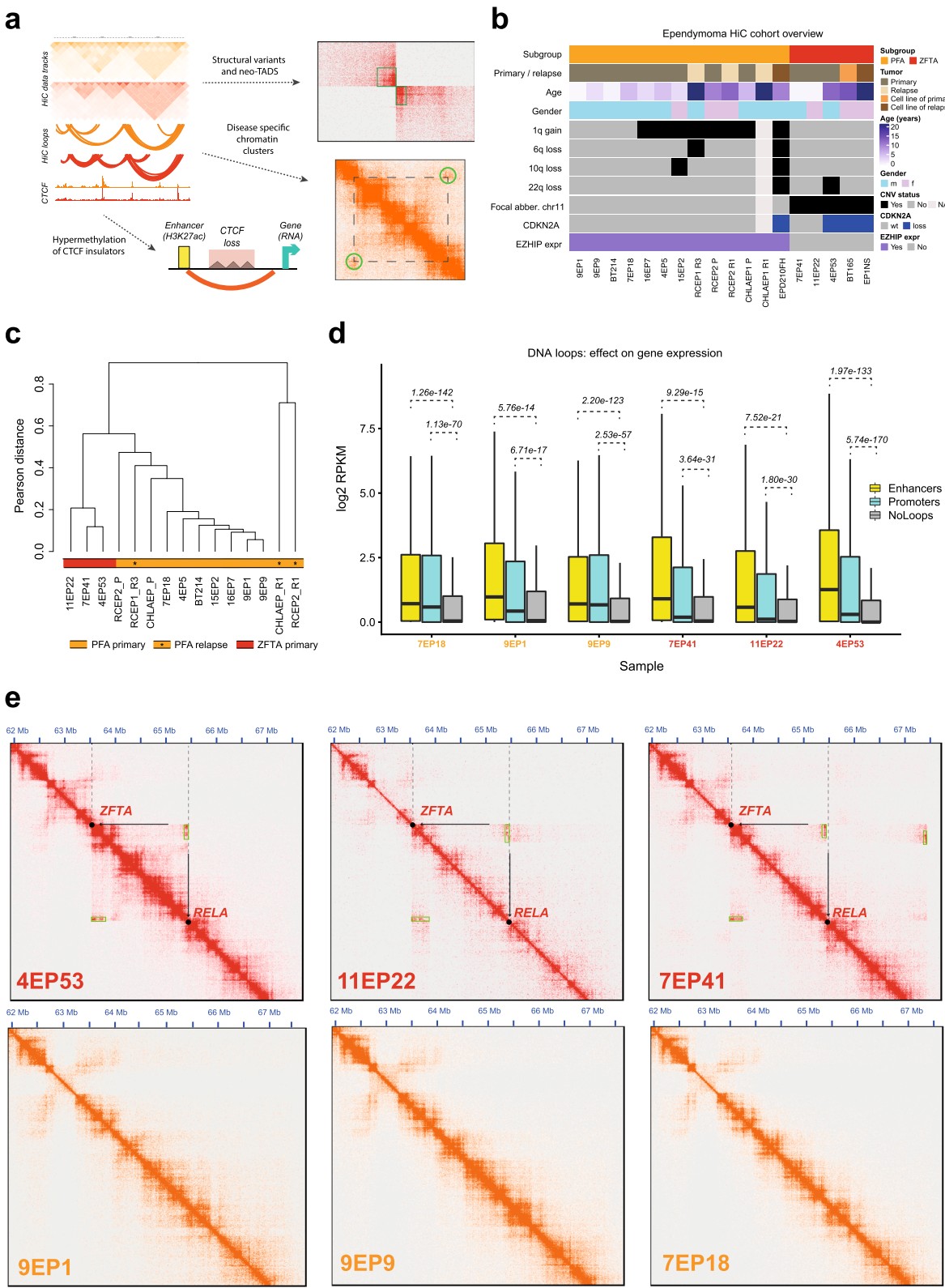

**Fig. 1 | 3D tumor genome profiling identifies PFA and ZFTA ependymoma specific chromatin conformations and enhancer associated genes. a** Overview of the major results obtained by the application of genome-wide chromosome conformation capture (Hi-C) in ependymoma brain tumors. **b** Characteristics of ependymoma samples analyzed by Hi-C. One group of PFA ependymoma samples has no apparent copy-number variants, while the other group of PFA samples exhibits chromosome 1q gains associated with an unfavorable outcome. In addition, the copy number status of selected chromosome arms that are most commonly affected in ependymoma is reported. **c** Unsupervised hierarchical clustering of PFA and RELA ependymoma tumors based on DNA interactions (Hi-C) stratifies the samples into the expected molecular groups. **d** Integrative analysis of enhancers (H3K27ac ChIP-seq), chromosome conformation (Hi-C) and gene expression (RNA-seq) shows that genes are more strongly expressed when their promoters physically interact with other promoters or with enhancers. Shown are tumors ($n = 6$ independent samples) for which sample-matched H3K27ac ChIP-seq, RNA-seq, and Hi-C data are available. *P*-values from the bootstrap t-test are included. The box plot center line, box limits and whiskers indicate the median, upper/lower quartiles and 1.5× interquartile range respectively. e) The Hi-C data reliably detect the structural variants that lead to the ZFTA-RELA fusion gene in supratentorial ZFTA-fusion associated tumors (top row), while no such signals were found in PFA tumors (bottom row). Green boxes highlight SVs predicted by the computational methods applied.

frozen or FFPE) and three cell lines (Fig. 1b, Supplementary Fig. 1a). A subset of samples was also analyzed by chromatin immunoprecipitation targeting the histone modification H3K27ac, which is associated with active chromatin, followed by sequencing (ChIP-seq, $n = 9$), transcriptome (RNA-seq, $n = 15$), whole genome sequencing (WGS, $n = 12$) and DNA methylation analysis ($n = 17$, Supplementary Data 1).

PFA and ZFTA ependymoma groups can be clearly distinguished using various molecular profiling techniques including DNA methylation[17] (Supplementary Fig. 1b). Unsupervised clustering of the Hi-C data clustered ependymoma tumors into the expected groups, demonstrating pronounced group-specific 3D tumor genome conformations (Fig. 1c, Supplementary Fig. 1c, d). Samples with a low number of valid Hi-C read pairs (<1e + 08, Supplementary Fig. 1a) were excluded for downstream analysis of DNA loops but could be used to identify larger structural variants (see below and Methods). By an integrative analysis of DNA loops derived from the Hi-C data with enhancers (defined by H3K27ac ChIP-seq enrichments), and gene expression (from RNA-seq profiles) for samples for which all three data types are available ($n = 6$), we observed that genes are generally expressed at higher levels when their promoters physically interact with enhancers or other gene promoters via the chromatin loops (Fig. 1d). Overall, we found that around twice as many genes as previously reported[18] are potentially regulated by proximal and distal ependymoma enhancers (Supplementary Fig. 1e, Supplementary Data 2).

We further applied computational tools for the detection of structural variants based on Hi-C data, which were previously applied for the identification of complex structural variants (SVs) in other cancer types[19,20], however, have not yet been applied to tumor biopsy samples (frozen and FFPE) from ependymoma patients. As a result, multiple SV candidates were uncovered from this analysis (Supplementary Data 3). We first took a closer look at the *ZFTA* and *RELA* gene loci, because it was previously shown that the oncogenic *ZFTA-RELA* gene fusions are a result of chromothriptic events on chromosome 11. As expected, the Hi-C data reproducibly detected SVs at the *ZFTA* and *RELA* gene loci in the supratentorial ZFTA but not in PFA tumors (Fig. 1e), which was verified by WGS and RNA-seq data (described in Methods). Furthermore, the Hi-C data captured extraordinarily complex rearrangements within chromosome 11 in some ZFTA ependymoma samples (Supplementary Fig. 1f) and revealed that SVs are not restricted to chromosome 11 but also include inter-chromosomal rearrangements (Supplementary Fig. 1g).

## Structural variants in supratentorial ZFTA ependymoma put RCOR2 into neo-TADs

The formation of new topologically associating domains ('neo-TADs') through structural variation was recently shown to have a critical role in gene dysregulation and oncogenesis[19,21]. To dissect the effect of structural variants in supratentorial ZFTA tumors on the potential formation of neo-TADs, we took a closer look at the *ZFTA* and *RELA* gene loci. By computational reconstruction of the tumor genome (Fig. 2a), we observed the formation of neo-TADs in all ZFTA samples,

placing the *REST Corepressor 2* (*RCOR2*) gene in new regulatory environments (Fig. 2b, c). *RCOR2* is located ~150 kb away of *ZFTA* and has a strong enhancer element upstream of its transcription start site that forms new DNA interactions with the *ZFTA* gene and other nearby enhancer elements by bridging the *ZFTA-RELA* breakpoint (Supplementary Fig. 2a). By evaluating Affymetrix gene expression array data[4] across ependymoma groups, we found that *RCOR2* expression is significantly upregulated in ZFTA relative to other ependymoma groups (*p*-value = 7.62e−27, Fig. 2d) and is highly correlated with *ZFTA* transcription ($R = 0.66$, *p*-value = 6.93−11, Fig. 2e). These results suggest that RCOR2 is a tumor-dependency gene in supratentorial ZFTA ependymoma tumors that may be transcriptionally activated by the SV-induced neo-TADs, but may also be lineage-specifically expressed or a direct target of the fusion gene[22,23]. To validate the relevance of RCOR2 for ependymoma cell growth, we performed shRNA-mediated knock-down of *RCOR2* expression in patient derived ZFTA and PFA cell lines. We observed that *RCOR2* knock-down results in strongly reduced cell growth of ZFTA cells and to a lesser extent of PFA cells (Fig. 2f, g, Supplementary Fig. 3a–c). The on-target effect of shRNAs against *RCOR2* was confirmed by western blot analysis in ZFTA and PFA cell lines (Supplementary Fig. 3d, e). Using an AnnexinV apoptosis assay we demonstrated induction of early apoptosis upon *RCOR2* knock-down only in ZFTA but not in PFA cells, indicating a group-specific function of *RCOR2* (Supplementary Fig. 3f, g). By gene expression analysis (Affymetrix arrays) upon *RCOR2* knock-down, we observed significant changes in neurogenesis related genes (e.g. *SFRP1, UCHL1, EDNRB,* Supplementary Fig. 2b) in PFA cells, whereas ZFTA cells showed significant changes mainly in apoptotic genes (e.g. *SRPX2, PERP, MZT1,* Supplementary Fig. 2c), suggesting a non-canonical function of *RCOR2* in ZFTA ependymomas[24,25].

*RCOR2* can form a protein complex with the histone demethylase *LSD1*, also known as *KDM1A*, and other transcriptional co-repressors, including *HDAC1/2*[24]. *LSD1*, *HDAC1* and *HDAC2* are all highly expressed across ependymoma groups, but *HDAC1/2* transcription is even more pronounced in ZFTA ependymoma (Supplementary Fig. 2d–f). In addition, *LSD1* and *RCOR2* transcription are strongly correlated in ZFTA but not in PFA ependymomas (Supplementary Fig. 2g). Since there is no available compound against RCOR2, we reasoned that inhibition of other components of the RCOR2/LSD1/HDAC complex may confer a therapeutic vulnerability for ZFTA ependymoma. shRNA-mediated inhibition of *LSD1* expression indeed leads to a significant depletion of ZFTA but not PFA cells compared to scrambled shRNA (Fig. 2h, i, Supplementary Fig. 3h, i). Surprisingly, however, targeting the enzymatic activity of *LSD1* with several *LSD1* inhibitors (ORY-1001, ORY-2001 and GSK2879552) had no effect on cell viability using clinically accessible concentrations (Fig. 2j, Supplementary Fig. 3j), suggesting that in this protein complex the protein rather than the enzymatic activity of *LSD1* is important. In contrast, targeting the HDAC activity with Entinostat, an *HDAC1-3* inhibitor, strongly inhibited the cell viability of ZFTA cells, while having less effect on PFA cells (Fig. 2k). Besides, the results for Corin, a dual inhibitor of both *LSD1* and *HDAC*s, were not better than those for Entinostat alone (Supplementary Fig. 3k).

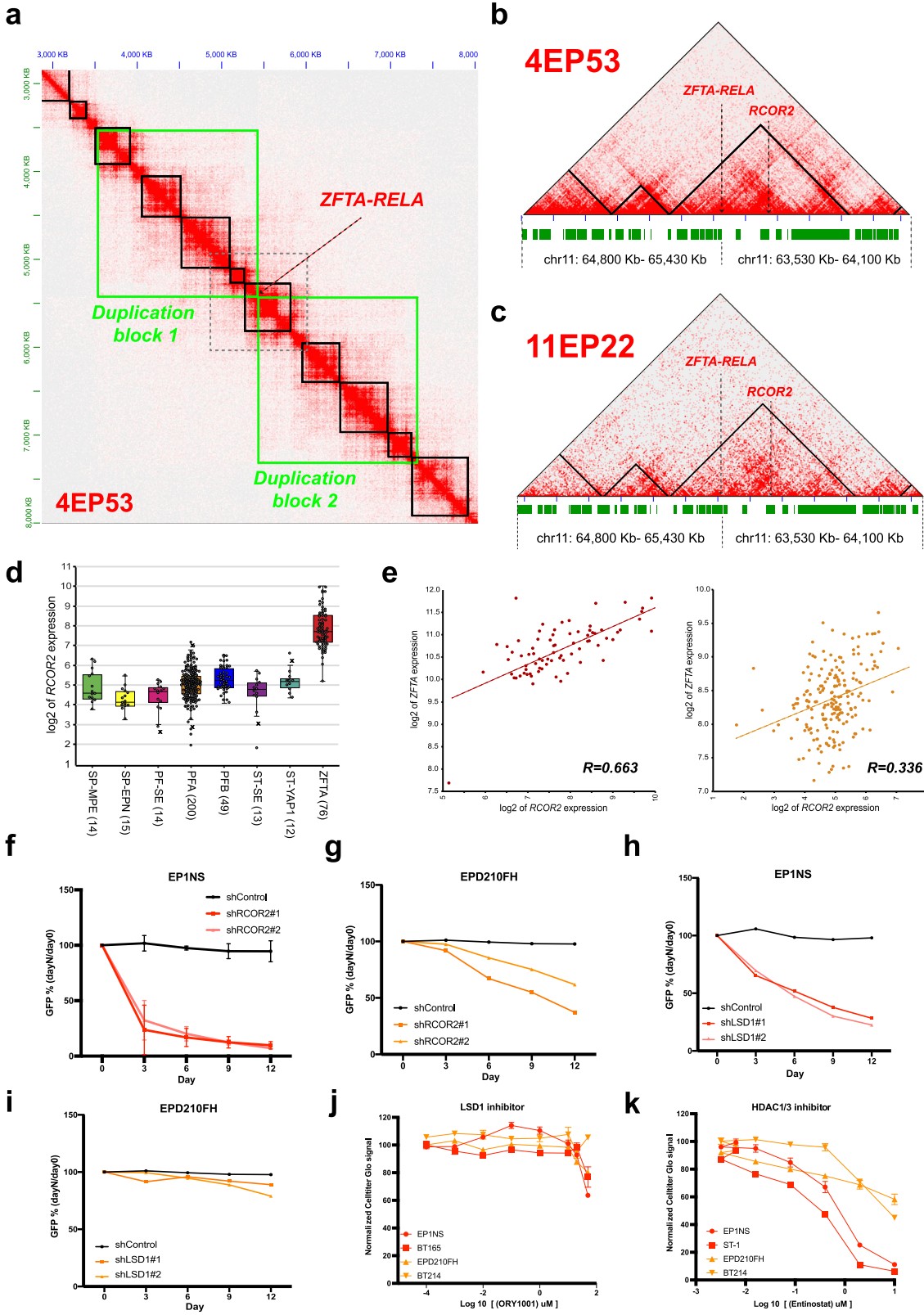

Inhibition of other HDACs with e.g. *HDAC8* and *HDAC6/10* inhibitors PCI-34051 and Tubastatin, respectively, had no effect on cell viability (Supplementary Fig. 3l, m). Altogether, our data show that the CoREST protein complex containing *RCOR2, LSD1*, and *HDAC1/2*, has a crucial role in the growth and maintenance of ZFTA cells that can be inhibited by disrupting the complex or by pharmacological targeting of *HDAC1/2*.

## Long-range DNA loops reveal a complex chromatin complex in PFA ependymomas

In all PFA tumors the Hi-C data revealed a 3D chromatin cluster that spatially links numerous regulatory sequences and genes located more than 4 million base pairs apart on chromosome 2 (Fig. 3a–c, Supplementary Fig. 4a). To determine if this chromatin cluster is specific to PFA tumors, we analyzed Hi-C data obtained from ZFTA ependymoma

**Fig. 2 | Transcriptional activation of RCOR2 by neo-TADs in RELA ependymoma. a** Chromatin contacts in a reconstructed ZFTA tumor genome (sample 4EP53) including the tandem duplication that leads to the ZFTA-RELA fusion (chr1:63532174-65429788, green boxes). Solid black boxes show TADs identified by applying TopDom to the Hi-C data mapped on to the reconstructed tumor genome, including a neo-TAD that spans the DNA breakpoint. **b**, **c** Reconstructed genomic locus containing the ZFTA-RELA fusion gene in the ZFTA ependymoma sample 4EP53 (**b**) and 11EP22 (**c**). The black boxes/ triangles indicate TADs reported by TopDom when applied to the reconstructed tumor genome. A neo-TAD is identified that spans the DNA breakpoint and places RCOR2 into a new regulatory environment. **d** Boxplot of RCOR2 gene expression across ependymoma groups using Affymetrix gene expression data ($n = 393$). RCOR2 is significantly upregulated in ZFTA tumors (ZFTA vs all other tumor classes limma $p$-val.: 7.62e−27). The center line, box limits, whiskers, and points indicate the median, upper/lower quartiles, 1.5× interquartile range and outliers, respectively. **e** Correlation between RCOR2 and ZFTA in ZFTA (left side, $n = 76$, cor = 0.663, $p$-val = 6.93e−11) and PFA

ependymoma samples (right side, $n = 200$, cor=0.336 $p$-val = 1.13e−06). **f−i** shRNA time-course knock-down experiments in ZFTA (EP1NS) and PFA (EPD210FH) ependymoma cell lines using a scrambled control and two shRNA constructs each targeting either RCOR2 in EP1NS (**f**), RCOR2 in EPD210FH (**g**), LSD1 in EP1NS (**h**) and LSD1 in EPD210FH (**i**). All constructs are GFP tagged and GFP positive cells are sorted by FACS. For panel (**f**), error bars represent mean ± SD for $n = 3$ independent experiments (two-tailed paired $t$ test $p$-val = 0.0018 and 0,0046; shRCOR#1 and shRCOR2#2, respectively). For panels (**g−i**), normalized data represent mean from $n = 2$ independent experiments per cell line. **j**, **k** Dose−response curves of single-compound treatment with ORY-1001 (**j**) or Entinostat (**k**) of ZFTA (EP1NS, BT165 and ST-1) and PFA (EPD210FH, BT214) ependymoma spheroids over a 72-h time-course using Celltiter-Glo cell viability assays. For each sample the results are presented as percentage of the Luminescence signal from control condition (i.e. water for ORY-1001 and DMSO for Entinostat). Error bars represent mean ± SD for $n = 3$ independent experiments (one-way ANOVA test $p$-val < 0,0001).

tumors as well as normal human tissues and cell types analyzed by the ENCODE and PsychENCODE consortia[26,27]. There was no sign of similar DNA interactions in the ZFTA and non-tumor samples, suggesting that this chromatin cluster is potentially characteristic of the cellular lineage of PFA ependymomas (Fig. 3a, b, Supplementary Fig. 4a). At the outer ends of this chromatin cluster are the homeobox transcription factors *DLX1/2* on one side and the *HOXD* gene cluster on the other. Examination of the transcription of these genes, which are ~4Mbp apart on linear DNA, shows a significant correlation in PFA (0.339, $p$-value: 1.103e−06) but not in ZFTA (0.147, $p$-value: 0.2) tumors. Based on these results, we hypothesized that this PFA-specific chromatin cluster might be associated with the transcriptional activation of genes on which PFA ependymoma tumors depend. By differential gene expression analysis against other ependymoma tumors, we find that the *mitogen-activated protein kinase 20 (MAP3K20)* and *integrin α6 (ITGA6)*, the latter encoding the receptor of the extracellular matrix protein laminin, which are both embedded in the chromatin cluster, are transcriptionally upregulated in PFA ependymomas (adj. $p$-values: 1.39e−22, 2.29e−06; Supplementary Fig. 4b, c). Genetic inhibition of *MAP3K20* by CRISPR-Cas9 resulted in decreased growth of PFA, but not ZFTA cell lines (Fig. 3d−f; Supplementary Fig. 4d). Moreover, pharmacological *MAP3K20* inhibition using M443[28] revealed higher sensitivity of PFA (IC50: 16,7 uM) compared to ZFTA cells (IC50: 37,5 uM) (Fig. 3g). Fluorescence activated cell sorting (FACS) shows that the slight but significant increase of *ITGA6* transcription translates to high ITGA6 protein abundance specifically in PFA compared with ZFTA ependymomas and normal human fetal neuronal stem cells (HF7450, Supplementary Fig. 4e). Gene expression analysis indicates that transcription of *integrin β4* (*ITGB4*), but not *β1* (*ITGB1*), is significantly upregulated in PFA compared to ZFTA ependymoma and normal brain samples (Supplementary Fig. 4g, h), suggesting that the integrin α6β4 heterodimer is the functional form relevant for PFA tumors. However, future studies will be necessary to determine the relevance of the integrin α6β4 heterodimer in this tumor type.

Importantly, gene-ontology analysis shows that ITGA6-associated gene sets, such as extracellular matrix organization and positive regulation of cell migration, are among the most highly enriched biological processes when comparing overall gene expression profiles of PFA to other ependymoma groups (Supplementary Data 4). Moreover, recent genome-wide CRISPR-Cas9 inhibition screens have revealed that *ITGA6* is specifically essential in PFA cell lines compared to glioblastoma (GBM) cell lines and fetal neural stem cells (fNSCs)[29,30] (Supplementary Fig. 4f). Based on these results, we hypothesized that ITGA6 is essential for the proliferation of PFA ependymoma in a disease sub-group specific manner. To test this hypothesis, we performed CRISPR-Cas9 mediated *ITGA6* knock-out in PFA cells using two different single guide RNA (sgRNA) sequences targeting *ITGA6*. As a result, we observed that PFA cells (Fig. 3h), but neither ZFTA cells (Fig. 3i) nor

GBM cells (Supplementary Fig. 4i), showed a decrease in cell growth over time.

## Structural variants in recurrent PFA ependymoma tumors with 1q gain place LAMC1 into neo-TADs

Conventional copy-number variation (CNV) analyses previously showed 1q copy-number gains in a subset of very aggressive and recurrent PFA ependymomas (Supplementary Fig. 5a)[10]. By investigating CNVs in PFA tumors, including primary and recurrent tumors of the same patient, we observed increases in genomic instability in recurrent tumors, where the 1q gain emerges during tumor progression (Supplementary Fig. 5b). To elucidate the molecular mechanisms associated with 1q gain, we systematically searched for SVs in all PFA ependymoma samples using the Hi-C data (Supplementary Data 3). As expected, primary PFA ependymomas have frequent DNA interactions within chromosomes ('cis') and no DNA interactions indicative of structural variants, (Supplementary Fig. 5c). However, we observed complex inter-chromosomal DNA ('trans') interactions indicative of structural variants in PFA ependymoma relapse samples (Supplementary Fig. 5d, e). We further inspected the regions of such translocations to find any commonly affected genes between PFA relapse samples. As a result, we observed recurring events that lead to re-arrangements of the *laminin subunit γ1 (LAMC1)* gene located on the chromosome arm 1q. For example, we observed inter-chromosomal translocations of chromosome arm 1q into chromosome 8 (chr1-chr8 in sample EPD210FH, Supplementary Fig. 5d, f), or into chromosome 3 (chr1-chr3 in sample RCEP1 R3, Supplementary Fig. 5e, g). Computational analysis of the Hi-C data of the two relapse tumors EPD210FH and RCEP1 demonstrated that these SVs lead to the formation of neo-TADs, which place *LAMC1* into new regulatory environments (shown for EPD210FH in Supplementary Fig. 5h). By evaluating gene expression of the PFA ependymoma tumors in our cohort, we found that *LAMC1* transcription is increased in the 1q+ PFA ependymoma relapse tumors as compared to primary PFA ependymoma tumors (Supplementary Fig. 6c).

Motivated by these results, we have identified patient-matched primary and relapse tumor tissues from two additional PFA ependymoma patients with 1q gain (RCEP2 and CHLAEP). Analysis of the relapse tumors by Hi-C revealed that LAMC1 was involved in inter-(RCEP2_R1, Supplementary Fig. 6a) or intra- (CHLAEP_R1, Supplementary Fig. 6b) chromosomal translocations, respectively. However, in these two PFA relapse tumors, the SV breakpoints are located further away from the LAMC1 locus, and there is no immediate remodeling of its regulatory environment. LAMC1 transcription is upregulated in both relapse tumors compared with the respective primary tumors (Supplementary Fig. 6d), however, this may be explained by the increased copy number of the 1q arm. To further investigate a potential transcriptional activation of LAMC1 by nearby SV breakpoints in a

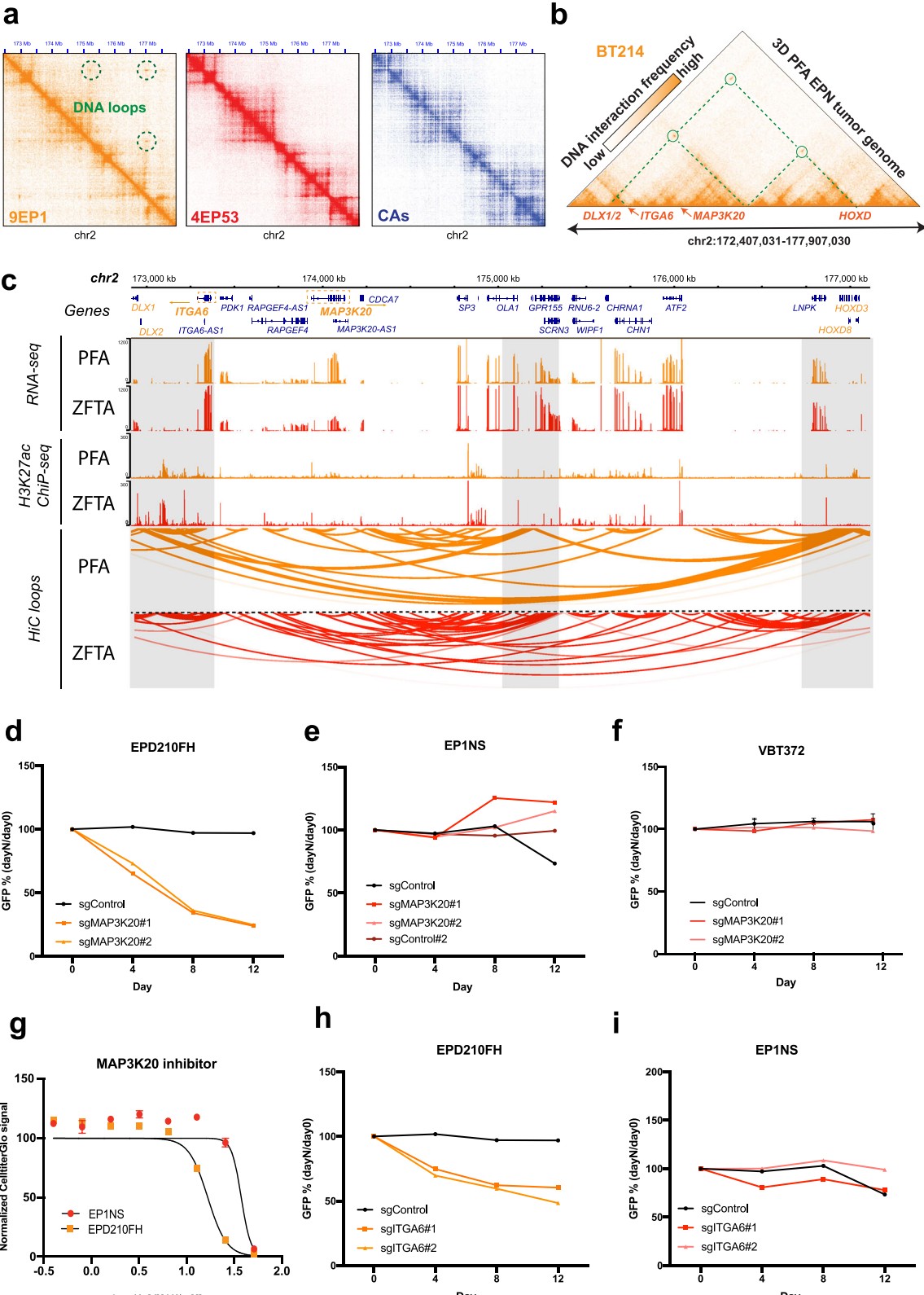

subset of PFA patients with 1q gain, we accessed an additional cohort of 29 PFA relapse patients from the INFORM project[31]. For the five PFA relapse samples with the highest *LAMC1* transcription in this independent cohort, we performed WGS and observed SVs involving *LAMC1* in the three cases that have the highest LAMC1 transcription (Supplementary Fig. 6e). Importantly, transcriptional activation was limited to *LAMC1* and does not affect nearby genes amplified together

with LAMC1 (Supplementary Fig. 6f). While transcriptional activation of LAMC1 by SV-induced neo-TADs may be a rare event and has not yet been functionally validated, we tested whether *LAMC1* is essential for cell growth of PFA ependymoma tumors and performed genetic inhibition experiments against *LAMC1* in the PFA cell line EPD210FH that harbors the chr1-chr8 translocation. As a result, we observed strongly reduced cell growth in PFA cells (Supplementary Fig. 6g, h), but not in

**Fig. 3 | Long-range DNA loops reveal a complex chromatin complex in PFA ependymomas. a** Hi-C DNA interaction matrices wherein a -5 million base pair segment of chromosome 2 is aligned along the diagonals shown for PFA (9EP1, left) and ZFTA (4EP53, middle) tumors and normal cerebellum astrocytes (CAs, right). **b** Hi-C DNA interactions of a PFA tumor (sample BT214) wherein the same -5 million base pair segment of chromosome 2 shown in panel (**a**) is aligned horizontally. Circles and dashed lines highlight long-range DNA interactions. **c** Genome browser view of the PFA-specific chromatin cluster shown in panels (**a**) and (**b**). The included data tracks show DNA interactions in PFA and ZFTA tumors via loops. Tracks for RNA-seq, H3K27ac and Hi-C derived DNA loop were obtained from merging PFA (9EP1,9EP9, 7EP18) or ZFTA (11EP22, 4EP53, 7EP41) samples, respectively. Differential specificity of the PFA loop is confirmed from statistical comparison (adjusted *p*-val 0.0089) via DiffLoop tool. **d**–**f** Genetic (CRISPR-Cas9) time-course inhibition

of MAP3K20 in one PFA EPD210FH (**d**) and ZFTA EP1NS (**e**) and VBT372 (**f**) cell lines. Changes in the percentage of GFP positive cells are presented after normalization. GFP percentage was normalized to day 4 post infection and presented as day 0. Normalized data represent mean from *n* = 2 independent experiments in cell lines EPD210FH, EP1NS and mean ± SD for *n* = 3 independent experiments for VBT373. **g** EPD210FH and EP1NS cells are treated with the MAP3K20 inhibitor (M443) for 6 days and cell viability is measured by CellTiterGlo assay and IC50 value is calculated by GraphPAD as respectively, 16, 7 and 37, 5 uM. Error bars represent mean ± SD, *n* = 3 biological replicates are used for all experiments. **h**, **i** Genetic (CRISPR-Cas9) time-course inhibition of ITGA6 in PFA EPD210FH (**h**) and RELA EP1NS (**i**) cells using a control sgRNA and two individual sgRNA constructs. All constructs are GFP tagged and GFP positive cells are sorted by FACS. Normalized data represent mean from *n* = 2 independent experiments in each cell line.

ZFTA (Supplementary Fig. 6i, j) or GBM cells as control (Supplementary Fig. 6k).

## Hypermethylation disrupts CTCF binding in PFA ependymoma

It has recently been shown that DNA methylation-mediated insulator dysfunction can lead to altered chromosomal topology, thereby activating oncogenic programs (Fig. 4a)[32,33]. Given the global loss of repressive H3K27me3[10,11] and a previously reported DNA methylation phenotype in PFA ependymomas[5], we hypothesized that similar molecular mechanisms may drive oncogenic transcriptional activation in this tumor type. Therefore, we analyzed seven PFA (*n* = 4) and ZFTA (*n* = 3) tumors using Whole Genome Bisulfite Sequencing (WGBS) and CTCF ChIP-seq (Supplementary Data 1). As expected, genome-wide CpG methylation is high in PFA and ZFTA ependymomas with low levels of methylation at functional regulatory elements, such as promoters, enhancers and insulators (Supplementary Fig. 7a). By comparative analysis of DNA methylation at CTCF binding sites, we found that DNA hypermethylation replaces 2,387 CTCF binding sites in PFA tumors, but conversely is associated with the replacement of only 178 CTCF binding sites in ZFTA tumors (Fig. 4b–d), indicating that the loss of CTCF binding through DNA hypermethylation is a predominant event in PFA ependymoma (Fig. 4c, Supplementary Fig. 7b).

To validate a potential interrelation of CTCF binding and DNA methylation in PFA, we treated PFA cells with the de-methylating agent 5-azacytidin. By using CTCF Cut&Tag, we observed genome-wide gain of CTCF binding sites in PFA cells upon induced de-methylation compared to PFA cells treated with DMSO (Supplementary Fig. 7c). Because loss of DNA methylation allows CTCF to bind at its DNA binding sites and thus resume its function as an insulator, we hypothesized that inhibition of DNA methylation simultaneously leads to loss of DNA loops. To test this hypothesis, we performed high-resolution promoter capture Hi-C in PFA cells treated with 5-azacytidine or DMSO, respectively. As expected, the results showed a strong loss of DNA loops in de-methylated compared to DMSO treated PFA cells (Supplementary Fig. 7d). Based on these results, we considered that DNA methylation-induced 'insulator dysfunction' leads to transcriptional activation of tumor-dependency genes in PFA ependymoma. To test this hypothesis, we first identified genes potentially activated in PFA by insulator dysfunction (Supplementary Data 5, see Methods). Among others, we observed localized hypermethylation in PFA tumors associated with the loss of a CTCF binding site and the formation of DNA interactions between enhancers and the *ADP Ribosylation Factor Like GTPase 4 C* (*ARL4C*) gene (Fig. 4e, f). *ARL4C* transcription is significantly upregulated (*p-value:* 1.63e−16) in PFA tumors compared to other ependymoma groups (Supplementary Fig. 7e) and is highly correlated with the activity of the enhancer elements that physically interact with the *ARL4C* gene locus in PFA but not in ZFTA tumors (Fig. 4g). It has been shown that *ARL4C* promotes migration, invasion and proliferation in colorectal and lung cancer[34], and recent genome-wide CRISPR-Cas9 inhibition screens revealed that *ARL4C* is essential for the proliferation of PFA ependymoma compared to

glioblastoma cell lines[29,30]. By genetic inhibition experiments we validated that *ARL4C* is indeed highly and specifically essential for the growth of PFA cells (Fig. 4h, Supplementary Fig. 7f), compared to ZFTA cells (Supplementary Fig. 6g–i) and glioblastoma cells (Supplementary Fig. 7h). To functionally validate the role of *insulator dysfunction* for transcriptional *ARL4C* activation, we designed two sgRNAs against the CTCF binding site that separates *ARL4C* from its associated enhancer in ZFTA tumors (Fig. 4f, i). As hypothesized, deletion of this CTCF insulator by CRISPR-Cas9 in a ZFTA cell line significantly increased *ARL4C* transcription (Fig. 4j). To further validate the role of DNA methylation-induced *insulator dysfunction* for transcriptional activation of other tumor-dependency genes in PFA tumors, we next focused on Negative Elongation Factor Complex Member B (*NELFB*) as this gene was previously identified as essential for PFA cell lines[29]. In PFA tumors we observed a physical interaction of *NELFB* with an ependymoma super enhancer mediated by a PFA-specific DNA loop. In ZFTA tumors, *NELFB* and the super enhancer are separated by a CTCF binding site that is replaced by DNA hypermethylation in PFA tumors (Supplementary Fig. 7j, k). By CRISPR-Cas9-mediated deletion of this CTCF binding site in a ZFTA cell line (Supplementary Fig. 7l–p), we observe significant increase of *NELFB* transcription (Supplementary Fig. 7n), supporting its role as regulatory insulator active in ZFTA but not in PFA ependymoma tumors.

## Discussion

By investigating 3D ependymoma genomes, we have identified multiple structural variants, chromatin conformations and tumor-dependency genes, pathways and potential therapeutic targets in ZFTA and PFA ependymomas. Since normal control samples for the different ependymoma groups are not well defined and inaccessible, our study largely focused on comparisons between different ependymoma groups. We show that structural variants in supratentorial ependymomas not only lead to *ZFTA-RELA* fusion genes, but also result in the formation of new regulatory environments that are recurrently associated with aberrant overexpression of *RCOR2*. RCOR2 is the scaffold protein in the CoREST complex that further contains LSD1 and HDAC1 and HDAC2. The complex is associated with gene silencing and is known to play a role in cancer development[35]. Here, we have shown that both *RCOR2* and *LSD1* expression are essential in ZFTA ependymoma, but not or to a lesser extent in PFA ependymoma, and that ZFTA cells are sensitive to HDAC1/2 inhibitors in line with our previous observations[36]. However, inhibition of the enzymatic activity of LSD1 had no effect. These results suggest that the activities of HDAC1/2 may be critical in regulating CoREST repressor functions in ZFTA ependymoma. Recent work in small cell lung cancer (and Merkel cell carcinoma) also implicated that disrupting the CoREST complex, but not the inhibition of LSD1's enzymatic activities, is required for blocking cancer cell proliferation[37]. Further studies identifying the components of the CoREST complex and identifying drugs that can disrupt the complex will be instrumental in developing an effective CoREST-targeted therapy for ZFTA ependymoma.

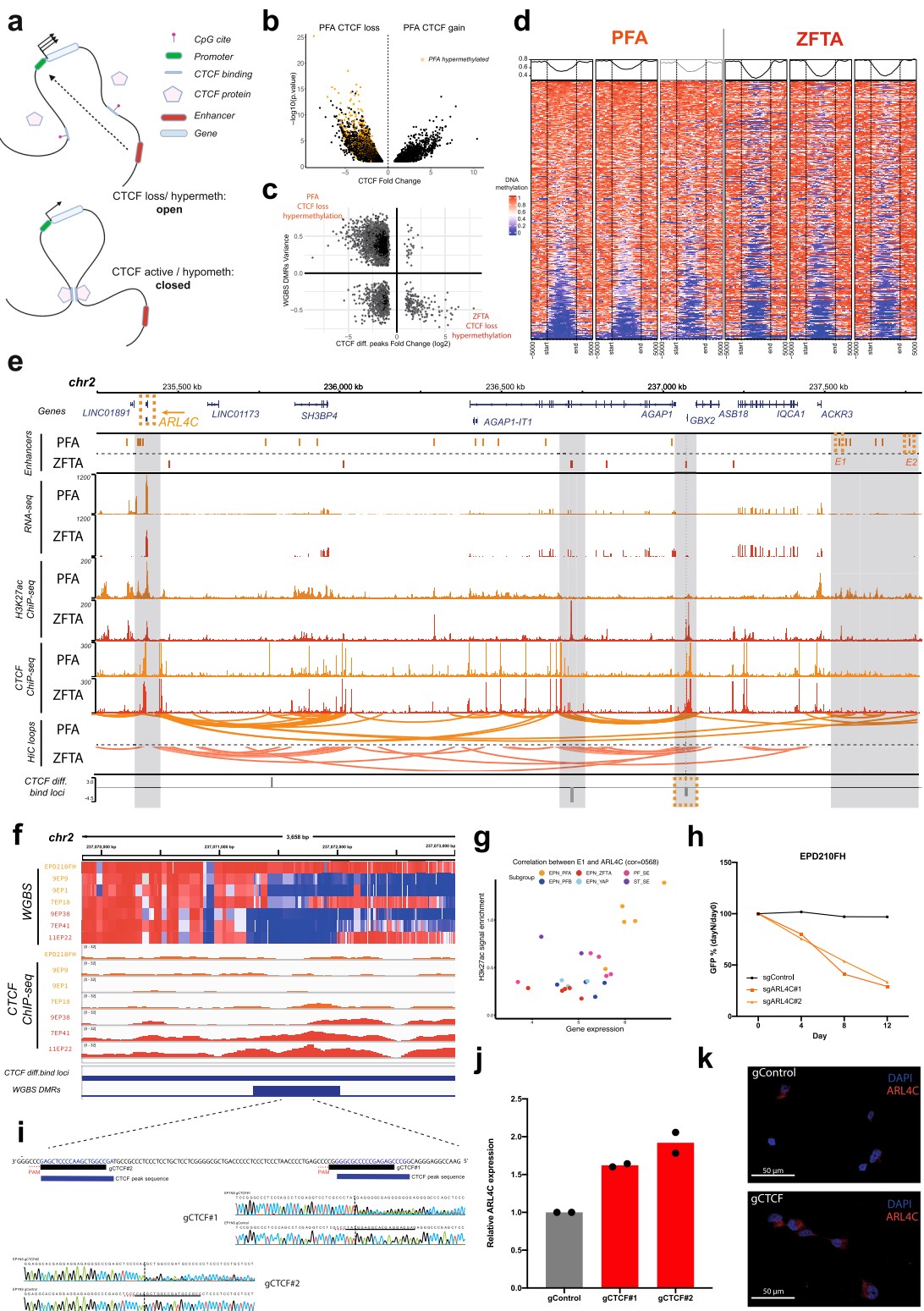

Furthermore, we have shown that PFA ependymomas are not only characterized by diminished histone methylation and increased acetylation at histone 3 lysine 27 (H3K27), as recently reported[29], but also exhibit characteristic 3D chromatin organizations. For example, the DLX1/2-HOXD chromatin cluster on chromosome 2 has specifically been observed in PFA ependymoma tumors and might be characteristic for the precursor cells of this tumor type as recently suggested[38].

Through targeting of *ITGA6*, a gene involved in this PFA-specific chromatin cluster, we demonstrated the importance of integrin signaling for maintained tumor growth, specifically in PFA tumors. *ITGA6* has been described as a marker for cancer stem cells (CSCs) in several cancer types[39-43], where disruption of ITGA6 function suppresses the CSC phenotype and the maintenance of stem cells[39]. Our results provide evidence for an epigenetic regulation event that could promote

**Fig. 4 | Hypermethylation replaces CTCF binding sites in PFA ependymoma.**
**a** Proposed mechanism of epigenetic oncogene activation in PFA ependymoma tumors. Top: The CTCF insulator is replaced by DNA methylation allowing enhancer to activate oncogene. Below: The oncogene is separated from an enhancer by topological barrier. Created with BioRender.com. **b** The volcano plot of differential CTCF binding sites between PFA and ZFTA ependymoma tumors (min p-value: 0.1). CTCF binding sites significantly hypermethylated in PFA are marked in orange (min q-value: 0.05). **c** Comparison of CTCF binding strength (CTCF ChIP-seq, x-axis, min p-value 0.1, min fold change: 0.5) and DNA methylation (WGBS, y-axis, min q-value: 0.05, min difference: 0.1) at differential CTCF binding sites between PFA and ZFTA ependymoma tumors. **d** Heatmap of WGBS-derived DNA methylation at the 300 most significant differential CTCF binding sites in three PFA (left) vs. three ZFTA (right) ependymoma tumors. **e** Genome browser visualization of PFA ependymoma-specific DNA loops that associate two PFA enhancers (E1 and E2) with the ARL4C gene. Tracks for RNA-seq, H3K27ac, CTCF and Hi-C derived DNA loops are obtained by merging PFA (9EP1,9EP9, 7EP18) or ZFTA (11EP22, 4EP53, 7EP41),

respectively. **f** WGBS-derived DNA methylation and CTCF ChIP-seq data from PFA and ZFTA tumors at PFA-specific hypermethylated CTCF loci. **g** ARL4C expression is positively correlated with activity of enhancer E1 (chr2:237763494 − 237764993) in ependymoma tumors (n = 24). **h** Genetic (CRISPR-Cas9) time-course inhibition of ARL4C in PFA cells (EPD210FH) using a control sgRNA and two individual sgRNA constructs. All constructs are GFP tagged and GFP positive cells are sorted by FACS. Normalized data represent mean from n = 2 independent experiments. **i** Expanded view of the CTCF motif targeted by CRISPR-Cas9: two sgRNAs and protospacer adjacent motif (PAM) direct Cas9 nuclease to the motif. Sequencing of target site demonstrates the formation of indels (insertion or deletions). **j** qPCR reveals increased ARL4C expression up on targeting CTCF by CRISPR-Cas9 in ZFTA cells. Results are normalized to control gRNA and data represent mean from n = 2 independent experiments. **k** Images depict ARL4C expression in ZFTA cells (EP1NS) after targeting the CTCF binding site by either gControl/Cas9 or gCTCF#1/Cas9 at 10 days post-infection.

integrin signaling in PFA ependymoma. The significance of integrin signaling for PFA tumor progression is further suggested by the recurrent transcriptional activation of *LAMC1* in PFA relapse tumors, which frequently harbor gains of chromosome 1q. Although relapse tumors often show increased genomic instability, our Hi-C data showed an unexpected complexity of intra- and inter-chromosomal rearrangements underlying some chromosome-arm-wide copy number variations in ependymomas. Our results suggest that transcriptional activation of *LAMC1* is a potential resistance mechanism in some recurrent 1q+ PFA EPN tumors that promotes proliferation by further enhancing already excessive integrin signaling. Notably, for other tumor types, LAMC1 has already been shown to be involved in tumor cell invasion and metastasis[44]. Thus, strategies that target integrin signaling, including ITGA6 and LAMC1, may reveal vulnerabilities and overcome resistance to therapy in the treatment of PFA EPN relapse patients.

Insulator dysfunction and oncogene activation by hypermethylation of CTCF binding sites has been described in IDH mutant gliomas and in SDH-deficient gastrointestinal stromal tumors (GISTs)[32,33]. Here, we show that PFA ependymoma is another tumor type with a global epigenetic phenotype in which there is hypermethylation of CTCF binding sites and associated changes in genome topology. By genetic inhibition of CTCF sites in ZFTA ependymoma lines, we provide evidence that insulator dysfunction is a potential oncogenic mechanism associated with transcriptional activation of the PFA tumor-dependency genes *ARL4C* and *NELFB*. Overall, our study has identified several group-specific tumor dependencies in ependymomas, opening avenues for potential therapeutic interventions that are urgently needed for this disease, especially for ZFTA and PFA ependymoma patients. We anticipate that our results will lay the foundation for further preclinical validation experiments in ependymoma patient-derived xenografts (PDX)[45] or in other in vivo models such as mouse models with ZFTA-RELA fusions[8]. Furthermore, our study shows that the analysis of 3D tumor genome may also be relevant for other (pediatric) cancers, for which the drivers are known but there are few therapeutic options.

## Methods
### Chromosome conformation capture
Hi-C on frozen tumor tissue sample was carried out using protocols previously described for tissue Hi-C experiments[46]. In brief, frozen tissues are pulverized using a mortar and pestle kept cold on a bed of dry ice into a fine powder. The tissue powder was then transferred to a 15 mL conical tube containing 5mLs of DPBS and fixed with 2% formaldehyde for 10 min. The fixation was quenched by addition of 0.2 M Glycine. The fixed tissue was pelleted by centrifugation, washed 1x with DPBS, and then flash frozen until ready for further processing.

For Hi-C experiments, the fixed frozen tissue pellets were first resuspended in 3mLs of lysis buffer (10 mM Tris-HCl pH 8.0, 5 mM CaCl$_2$, 3 mM MgAc, 2 mM EDTA, 0.2 mM EGTA, 1 mM DTT, 0.1 mM PMSF, 1X Complete Protease Inhibitors). The sample was transferred to an M-tube and dissociated using a GentleMACS Tissue dissociator (Miltenyi) using the "Protein M-tube" setting. The sample was removed from the M-tube into a 50 mL conical. The M-tube was washed with 3mLs of lysis buffer with 0.4% Triton X-100 added, and this wash was combined with the original 3mLs of sample for a total volume of 6mLs with final concentration of 0.2% Triton X-100. The sample was then passed through a 40 μM cell strainer. The strainer was washed with an additional 2mLs of lysis buffer with 0.2% Triton X-100. The sample was then centrifuged and washed with 1 mL of lysis buffer with 0.2% Triton X-100. After centrifugation, the sample was resuspended in 0.5% SDS and processed with previously described in situ Hi-C method[47] using the MboI enzyme. Libraries were prepared using the Illumina TruSeq LT sequencing adaptors. Initial QC sequencing was first performed on a MiSeq to assess library quality, and if sufficient, was subject to production scale sequencing on the HiSeq X or NovaSeq platform, respectively.

### Chromosome conformation capture from FFPE material
Hi-C experiments on FFPE material for samples RCEP1-R3 and CHLAEP-R1 were carried out by Arima Genomics, Inc (San Diego, CA). Dewaxed and re-hydrated FFPE tissue was used as input to a modified version of the Arima-HiC Kit protocol. After the Arima-HiC protocol, Illumina-compatible sequencing libraries were prepared by shearing the proximally ligated DNA and then size-selecting DNA fragments using SPRI beads. The size-selected fragments containing ligation junctions were enriched using Enrichment Beads (provided in the Arima-HiC Kit), and converted into Illumina-compatible sequencing libraries using the Swift Accel-NGS 2S Plus kit (P/N: 21024) reagents. After adapter ligation, DNA was PCR amplified and purified using SPRI beads. The purified DNA underwent standard QC (qPCR and Bioanalyzer) and sequenced on the NovaSeq following manufacturer's protocols.

### Hi-C data processing
The sequencing reads alignment to hg19 human genome reference and chromatin contacts calling was performed using HiCPro 2.9.0 toolkit[48] allowing the bin sizes 5,10,50,100,250, and 500 Kbp. Main visualization and coverage normalized full contacts extraction was performed with JuiceBox v0.7.5 toolkit[49]. Per sample loop calling was applied from FitHiC v2.0.6 method[50] on bin sizes 5 Kbp with maximum distance between bins 50 Mbp. TAD calling was performed based on 50 Kbp bins resolution using TopDom tool[51]. Differential DNA loop calling was performed based on a 5 kb resolution using the DiffLoop tool[52] and by applying an adjusted p-value 0.05.

## Unsupervised clustering of Hi-C data

Hi-C data were processed using Genome Contact Map Explorer[53]. Contact matrices were normalized using the iterative correction (IC) algorithm and binned into 500kbp windows, generating 457,851 features. Features were quantile normalized using preprocessCore v1.56.0. For clustering, the 20,000 features with the greatest variance were used. Hierarchical clustering was performed using "Pearson distance" ($1 − Pearson\ correlation$) and Ward (Ward.D2) agglomeration, using the *hclust* function in R 4.1.2 and visualized with dendextend v1.15.2.

## Identification of enhancer-associated genes (EAGs)

The ChIP-seq derived enhancer signals along with genomic locations of group-specific enhancers and normalized RNA-seq gene expression profiles from ependymoma tumors cohort ($n = 25$) were obtained from published materials of the corresponding study[18]. Genome was fragmented into 5 Kbp bins and output from FitHiC loop calling tool was used to find contacts between genes and enhancers. For this purpose the genes were assigned to bins based on the location of transcription start site (TSS, 2500 Kbp upstream and downstream of the gene stat loci), while enhancers based on the overlap. To search for connections between genes and enhancers, the FitHiC loops per sample were filtered based on minimum $p$-value 0.01. Further filtered loops from 3 PFA (9EP1, 9EP9, 7EP18) and RELA (7EP41, 11EP22, 4EP53) samples overlapping with the corresponding enhancer cohort were merged using the following rule: if the same loop repeats across several target group samples then only one loop with maximum statistical evidence among these samples is selected to represent the group, remaining unique loops are included without any changes. Loop boundary correspondence was assigned to gene and/or group-specific enhancer lying either within the bin or in the closest upstream/downstream bin. Enhancer-associated gene was considered to be supported by loop if the TSS of it was lying in one loop anchor while enhancer in the other. Comparison of genes expression in connection via loops to other genes, enhancers and unconnected genes was performed via bootstrap $t$ test: the set of genes not in connection to enhancers was selected randomly to reflect the number of genes as in connection (2–3% from total number of genes), this procedure was repeated 1000 times. Correlation analysis was performed based on the usage of updated InTAD package[54] v1.9.2. Shortly, for all detected gene-enhancer pairs connected via loop the correlation was computed using data from full enhancer cohort ($n = 25$). Further the minimum correlation limit 0.5 was applied for the filtering. Additional sources of information such as gene expression specificity for target tumor class in the enhancer cohort ($n = 25$) and global Affymetrix data ($n = 618$), loop presence in other group and normal cerebellum astrocyte, expression of gene across PFA and RELA cell lines in RPKM were included in the results tables (Supplementary Data 2).

## Gene expression analysis

The global ependymoma tumor gene expression data integration was performed based on the usage of corresponding R2 platform materials with focus on Affymetrix dataset from combined ependymoma tumors cohort with integration of normal brain tissues ($n = 618$). Major of these ependymoma tumor Affymetrix materials were obtained from the corresponding main study[4] (GEO: GSE64415) with additional external inclusions (GEO: GSE50161, GSE50385, GSE21687, GSE3526). The gene ontology analysis was performed using DAVID tool[55] based on the usage of differentially expressed genes between PFA/RELA and other ependymoma groups achieved with R2 platform from the EPN global Affymetrix dataset based on the usage of limma package[56]. The RNA-sequencing materials from target EPN cohort samples were analyzed as previously described[18].

## Analysis of structural variants (SV) using Hi-C data

SV discovery from Hi-C data was performed using two independent toolkits. The first toolkit, hicBreakFinder (https://github.com/dixonlab/hic_breakfinder), was adjusted for the usage on hg19 human genome reference with taking into account additional filtering lists of false positives obtained from external cohorts[19]. Shortly, the tool scans for abrupt shifts in chromosomal connections in order to find possible outliers representing inter/intra-chromosomal events based on the selected threshold ($t = 0.6$) and reports them in resolutions 1 Mb, 100Kb and 10Kb. Final combined result contains the highest resolution for detected SV. The second toolkit, Hi-C structural variant discovery or HiCsv, consists of two parts and was adjusted for the usage of hg38 genome as the most up-to-date reference genome. First part of this toolkit, HiCtrans[20], focuses on inter-chromosomal translocations: it scans the inter-chromosomal contact matrices over multiple Hi-C resolutions for each possible pair of chromosomes from a given sample and predicts candidate SVs based on the changepoint analysis using binary segmentation. The intra-chromosomal translocations are also detected in this toolkit based on the dual pattern of off-diagonal enrichment and diagonal depletion of chromatin interactions in a Hi-C map across genomic regions. HiCsv detects enrichment of interactions through FitHiC2 algorithm[50] and uses an insulation score-based estimation (similar to TAD finding[57]) to identify depletion in interaction frequency. Finally, it applies a density-based clustering of enriched Hi-C interactions with high insulation scores to discover structural variants.

In order to verify the SV discovery from Hi-C data we also integrated WGS ($n = 4$) and RNA-seq data ($n = 5$) available for RELA samples. By performing WGS SV calling with Delly[58] and RNA-seq fusion calling with InFusion[59] the ZFTA-RELA was recovered in all samples confirming obtained Hi-C SV results. Same verification of inter-chromosomal SV based on WGS data was performed for EPD210FH and FFPE relapse samples, confirming corresponding SVs. Genes possibly affected by SV were detected by expanding the borders of detected breakpoint up to 500 Kbp. Further extended SV genomic locations between EPD210FH and FFPE relapse samples were compared and 556 genes were found lying within the overlapping segments between them. Filtering of genes was performed based on differential expression specificity for PFA vs other tumor types from Affymetrix large cohort, RNA-seq enhancer cohort and also in comparison vs normal brain. LAMC1 was the only gene passing all filtering limits and lying on 1q chromosome arm. For verification of 1q gain effects in relapse tumors additional control bulk RNA-seq ($n = 29$) and WGS ($n = 5$) cohorts with processed materials (gene expression counts, structural variant calls) were produced and kindly provided by the INFORM program[31].

## Reconstruction of intra-chromosomal SVs at the *ZFTA-RELA* fusion gene

The reference blocks forming ZFTA-RELA fusion for samples 11EP22 and 4EP53 were reconstructed from WGS data using genomic duplication coordinates obtained using Delly SV calling tool (11EP22 - *11:63532555-65430159*, 4EP53 - *11:63532174-65429788*). Novel reference segment was formed starting from *chr11* starting at *60000000* and ending at *69000000* with duplications included within. All the contacts from HiC-Pro output lying in this region were converted to novel genomic coordinates with corresponding duplicate segments repeated. Further, since both normal and somatic allele were present in both RELA samples, we counted overlapping reads supporting duplication breakpoint and normal expected formation, resulting in proportion 50%: 50% to obtain correct proportional contact variance for duplication segment. Novel-formed contacts were used as input for TopDom for TAD calling and JuiceBox visualization generation.

## Reconstruction of inter-chromosomal SVs at the *LAMC1* locus

To confirm the neo-TAD formation with *LAMC1* gene in EPD210FH cell line we focused on reconstruction of the reference sequence to redo TAD calling. For this purpose, we checked the LAMC1 covering SV genomic loci (first segment in *chr8-80310000:80430000* and second segment in *chr1:182770000-184100000)* within WGS data to identify the corresponding precise breakpoint genomic coordinates in EPD210FH using Delly SV calling tool that detected inter-chromosomal SV *chr8:80427007−chr1:182968878*. Further, using this information we created a novel reference sequence formed from these chromosomal segments and converted Hi-C read contacts from HiCPro analysis results overlapping these regions to the novel coordinates. During the extraction of overlapping Hi-C contacts we also took into account allele variance in the support of SV for correct adjustment of contact proportions since the normal allele was also present in the WGS data (60% with somatic vs 40% normal). In result we achieved a block of Hi-C data with novel reference. These Hi-C read contacts were used as input for TopDom tool to call TADs and JuiceBox tool for visualization as described previously.

## Promoter capture Hi-C of 5-Aza and DMSO treated PFA ependymoma cells

PFA cells (EPD210FH) were treated with either DMSO or 5 µM of 5-Azacitidine (5-Aza) from Selleckchem for 8 days in the cell culture conditions as described in Cell Culture method. Cell culture media was replenished with fresh 5-Aza or DMSO every third day. At the end of treatment, cells were harvested by Accutase treatment followed up centrifugation at 300 *g* for 5 min. Cell pellets were resuspended in 15 ml falcon tubes with room temperature (RT) 1x PBS solution and viable cell count was determined by Trypanblue staining. ~5 million cells for each treatment were taken and total volume filled up to 5 ml with adding required amount of RT 1x PBS. To crosslink the cells with 2% formaldehyde, 268 µL of 37% formaldehyde (SantaCruz) was added into each falcon tubes. Samples were mixed well by inverting tubes ten times and incubated later at RT for 10 min. Then, 460 µL of 2.5 M glycine (Sigma) added on top and after mixing well by inverting the tubes, samples were incubated at RT for 5 min. Samples further were incubated on ice for 15 min and pelleted by centrifugation at 500 *g* for 5 min. Samples were resuspended with 5 ml of 1x PBS to obtain 1 million cells per ml. 1 ml of samples were transferred into 1.5 ml ependorf tubes and centrifuged for 5 min at 500 *g* to form pellets. Pellets were kept at −80ºC until performing Hi-C preparation.

Promoter capture Hi-C was performed was performed using the Arima-HiC+ promoter cHiC kit (Arima Genomics, Inc., Cat # A301010), comprising a capture panel targeting the promoter regions of 23,711 human genes. Main reads processing was performed with HiCUP pipeline[60] adjusted for the protocol. Significant interactions (loops) were identified with CHiCAGO toolkit[61] based on default setting except subset of parameters (*minFragLen: 4000, maxFragLen: 6000, binsize: 25000, minNPerBait: 250, maxLBrownEst: 2000000*). Differential loops between 5-Aza and DMSO were analyzed using the diffloop[52] package.

## CTCF ChIP-sequencing

ChIP-sequencing procedure was prepared and performed as previously described[18]. Shortly, ChIP flash-frozen for ependymoma tumors was performed using 5 µg CTCF antibody per ChIP (Active Motif #39357). Enriched DNA was quantified and barcoded. Following library amplification, DNA fragments were sequenced using Illumina HiSeq 2000 100-bp paired-end sequencing.

## CTCF ChIP-seq data analysis

Reads alignment was performed to hg19 reference with BWA v0.5.10[62]. Duplicate alignments were removed using Picard (http://broadinstitute.github.io/picard). Peak calling was performed using Macs v1.4[63]. Differential RELA peaks between EPN PFA and RELA were detected using DiffBind R package[64] with min adjusted *p*-value limit 0.05 resulting in 2436 peaks.

## CTCF Cut&Tag of 5-Aza and DMSO treated PFA ependymoma cells

PFA cells (EPD210FH) were treated with either DMSO or 5 µM of 5-Azacitidine (5-Aza) from Selleckchem for 8 days in the cell culture conditions as described in Cell Culture method. Cell culture media was replenished with fresh 5-Aza or DMSO every third day. At the end of treatment, cells were harvested by Accutase treatment followed by centrifugation at 300 *g* for 5 min. Cells were further prepared according to ActiveMotif protocol (https://www.activemotif.com/documents/2205.pdf) and CTCF Cut&Tag was performed as a service by ActiveMotif.

## CTCF Cut&Tag data analysis

The analysis was performed as for ChIP-seq data processing. Differential RELA peaks between 5-Aza and DMSO were detected using DiffBind R package[64] with min adjusted *p*-value limit 0.05 resulting in 1126 peaks.

## Whole genome bisulfite sequencing (WGBS)

WGBS procedure was prepared and performed as previously described[65]. Shortly, 5 µg of genomic DNA were sheared using a Covaris device. After adaptor ligation, DNA fragments were isolated and bisulphite converted using the EZ DNA Methylation kit (Zymo Research). PCR amplification of the fragments was performed followed by library aliquots pooling. Sequencing was performed Illumina HiSeq 2000 machine.

## WGBS data analysis and its combination with CTCF peaks

Initial reads processing was performed using methylCtools v0.9.4 as previously described[65]. Differentially methylated regions (DMRs) were detected using metilene v0.2.6 tool[66] with min adjusted *p*-value limit 0.05 resulting in 9384 regions. Combined visualization of the methylation profiles within CTCF target regions was performed using the EnrichedHeatmap R package.

The DMRs were overlapped with CTCF differential peaks (min adj.*p*-value 0.05) resulting in 1254 pairs. The analysis for gene selection was performed on hypermethylated CTCF loci for PFA (*n* = 966) and ZFTA (*n* = 43) using two different approaches. From both of them no results were found for ZFTA, but only for PFA.

In the first method the target loci were checked to be lying within the loops between associated genes and enhancers detected previously (Supplementary Data 2). For selection of enhancer associated genes additional filtering criteria were applied: gene is specific for target group based on differential expression comparison within enhancer RNA-seq cohort (*n* = 25) and global Affymterix (*n* = 618), minimum expression in cell lines−1 RPKM. The filtering resulted in 510 gene connections (Supplementary Data 5a).

In other method, association of super enhancers (SE) with target genes via lost in PFA CTCF hypermethylation sites (*n* = 966) was performed by integrating full SE list from the EPN corresponding study computed via InTAD package[54] v1.9.2. The loops starting with one edge from SE and covering initially DMRs overlapping with differential CTCF site were used to find connected genes significantly differentially expressed in PFA vs RELA tumors in global Affymetrix cohort (*n* = 618), resulting in 476 candidates (Supplementary Data 5b). To strengthen the specificity additional information was included to mark genes that were previously verified to be PFA specific via CRISPR-Cas vs fNSC and GBM cell lines.

## Preparation of genome browser tracks

For the visualization of selected genomic loci via Integrative Genomics Viewer[67], BAM files from samples of the same ependymoma group

were merged using samtools and further converted into bigWig files using UCSC tools. The loops visualization tracks were generated by combining loops from 3 PFA and 3 RELA samples that have a minimum *q*-value of 0.05.

## Cell culture

HEK293T cells (CRL-1273, American Type Culture Collection) were cultured in DMEM-Glutamax (Gibco) medium supplemented with 10% FBS (Gibco) and 1% Penicillin/Streptomycin (Gibco). EPD210FH and BT-214 cells were grown in NeuroCult NS-A Basal Medium (STEMCELL Technologies) supplemented with NeuroCult Proliferation Supplement (STEMCELL Technologies), 2mM L-glutamine 1% Penicillin/Streptomycin, 75 ng/ml bovine serum albumin (BSA) and 20 ng/ml of EGF (Peprotech) and FGF-basic (Peprotech). EP1NS, BT-165, ST-1 and VBT372 cells were grown in Neurobasalmedium A (Life Technologies) supplemented with 1 µg/ml of Heparin (Sigma), 2mM L-Glutamine, B27 without Vitamine A (Life Technologies) and 20 ng/ml of EGF and FGF-basic. Cells were cultured as neurospheres in tissue culture flasks. When they were cultured as an adherent culture, flasks were additionally coated with Laminin (L2020, Sigma) for EPD210FH, BT-214 and ST-1 cells and with Geltrex (A1569601, Thermo Fisher) for all other ependymoma cells. Pediatric patient-derived GBM2 cells were cultivated as neurospheres as previously described[68]. GBM2 cells were grown in a mixed DMEM/F12 (Life Technologies) and Neurobasal(Gibco) medium supplemented with following components 1 M of Hepes, Sodium pyruvate, MEM, PDGF-AA growth factor in addition to 1 µg/ml of Heparin (Sigma), 2mM L-Glutamine, B27 without Vitamine A (Life Technologies)and 20 ng/ml of EGF and FGF-basic. All cells were routinely tested for mycoplasma contamination and authenticated by Single Nucleotide Polymorphism profiling (Multiplexion GmbH). All cell models were grown at 37 °C with 5% CO2. Catalog numbers of medium components in details can be found in Supplementary Data 9.

## Lentiviral plasmids used in the study

pL.CRISPR-GFP and pL.CRISPR-puro plasmids were a kind gift from Jovan Mirčetić and pRSI12-TagGFP and pRSI12-TagRFP plasmids were bought from Cellecta. To clone shRNAs in to pRSI12 and gRNAs to pL.CRISPR, two complementary oligos with required overhangs (see Supplementary Data 6) were annealed and phosphorylated by mixing 1 µL of each oligo (at 100 µM) with 1 µL T4 ligation buffer (Thermo Fisher) and 0.5 µL polynucleotide kinase (Thermo Fisher) in a 10 µL reaction. The reaction was run at 37 °C for 30 min, followed by 95 °C for 5 min and slow cooling to 25 °C (0.1 °C/s). Golden gate cloning was performed afterwards: 100 ng lentiviral vector was mixed with 1x Tango buffer (Thermo Fisher), 50 nM DTT, 0.5 µL T4 ligase (New England Biolabs), 1 µL Esp3 (Thermo Fisher) and 2 µL 250x-diluted annealed oligos in a 20 µL final reaction. The reaction was run for 6 cycles that consisted of 5 min at 37 °C/5 min at 20 °C. The reaction was transformed into Stbl3 chemically competent *E.coli* cells and individual colonies were picked. The gRNA or shRNA identity was confirmed by Sanger sequencing using U6 or Ubic promoter primers respectively. All oligos were ordered from Sigma. Oligo sequences with overhangs as well as primer sequences are listed in Supplementary Data 6.

## Competitive growth assay with FACS

Ependymoma cells were expanded as described above and harvested by Accutase treatment (Sigma-Aldrich). Then, 5 µg/ml protamine sulfate (Sigma-Aldrich), virus for shRNA or sgRNA expressing constructs re-suspended in the respected medium together with (0.2–0.3) × 10⁶ cells were seeded in to coated 24-well cell culture plates and then incubated at 37 °C overnight. Lentivirus for gRNA constructs were concentrated up to 10 times by using Lenti-X (Takara) before using. Transduced cells were split three days post-infection and further kept in 96-well flat bottom coated cell culture plates for the detection of GFP or RFP by Fluorescence-activated cell sorting (FACS) using a BD

Fortessa or BD Canto. The first FACS measurement was done one day after seeding (4 days post-infection−reference sample for competition assay, termed Day0). FACS measurement was performed every 3–4 days using one-two wells for each time point. Data were further processed using FlowJo software (version 10.7 and 10.8). In detail, at least 5000 single cells were analyzed. To gate the cells, forward versus side scatter area (FSC-A/SSC-A) was used to identify live cell population and the doublets were excluded using the FSC height versus FSC area plot (FSC-H/FSC-A). Single cells were plotted in FSC area versus FITC area channel (FSC-A/FITC-A) and percentage of GFP or RFP-positive cells was determined. Percentage of GFP or RFP-positive cells at Day0 was scaled to 100% and every subsequent measurement was normalized to this value. The relative proliferation of GFP+ or RFP+ cells within the total cell population was monitored by FACS and indicated in the manuscript as cell growth. Statistical analysis was performed using Prism8 (GraphPad) for all experiments wherever it is applicable (3 independent experiments) but only significant ones were mentioned in Figure legends.

## Drug treatments

All drugs were prepared according to protocols provided by the companies (Supplementary Data 7). Cells were seeded into 96-well cell culture treated plates at a density of 5000 cells in 100 µl respected medium per well. Sixteen hours after plating, cells were treated with increasing concentrations (variable for different drugs. See Source data for exact concentration) of each drug or equivalent dilutions of solvent and cell viability was assessed after 3 or 6 days using the CellTiter-Glo luminescent cell viability assay (Promega) and an automated plate reader Mithras (Berthold). All samples were assayed in technical triplicates and normalized to the average values of the corresponding solvent control on the same plate. The same experiments were repeated three times independently and analyzed using Prism 8 (GraphPad Prism). Statistical analysis was performed for all experiments but only significant ones were mentioned in Figure legends.

## Western blot analysis

For knockdown and/or knockout studies, cells were infected as described above and cultured for either 4 days (knockdown) or 5 days (knockout). Before harvesting, cells were washed with phosphate buffered saline (PBS) and collected as pellets. Then, pellets were lysed in RIPA buffer (Sigma-Aldrich) supplemented with protease and phosphatase inhibitors (Roche) for 30 min on ice. After centrifugation at 10,000 × *g* for 10 min at 4 °C, the supernatants were collected and protein concentrations were determined using a Bicinchoninic acid (BCA) assay (Pierce, Thermo Fisher) Lysates were mixed with NuPAGE® LSD Sample buffer (Life Technologies) supplemented with 10% 2-mercaptoethanol and denatured for 5 min at 95 °C. Afterwards, they were subjected to sodium dodecyl sulfate-polyacrylamide gel electrophoresis according to standard procedures using 4–12% Bis-Tris gels (Thermo Fisher) and afterwards transferred to polyvinylidene difluoride membranes. Membranes were incubated with respective primary antibodies at 4 °C overnight (Supplementary Data 8). Horseradish peroxidase-conjugated anti-rabbit secondary antibody (Cell Signaling, 1:2500 dilution) was applied for 1 h at room temperature and chemiluminescent detection was carried out using Amersham™ ECL™ or ECL™ Prime Western Blotting detection reagents (GE Healthcare). The same membranes were stripped with stripping buffer (Thermo Fisher) according to protocol and incubated with conjugated beta-actin antibody (Abcam, 1:10000 dilution, #ab49900) as a loading control. All uncropped scans can be found in Source Data.

## Immunostaining

Coverslips were placed into 48-well cell culture plates under aseptic conditions and were coated with Geltrex (Life Technologies). Ten days after infection, transduced cells were grown overnight on these

coverslips at 37 °C for overnight. Afterwards, cells on coverslip were fixed with ice-cold 4% PFA (Santa Cruz), then washed twice with 2 mM $MgCl_2$ in PBS for 5 min each. Cells were then permeabilized with 0.1% Triton X-100 (Sigma) in PBS for 10 min at room temperature, followed by two washes with PBS. Subsequently, cells were blocked in blocking solution (1x PBS, 5% donkey serum Sigma-Aldrich) for 15–30 min at room temperature. The primary antibody was diluted 1:200 in blocking solution and incubated 1 h at room temperature (rabbit anti-ARL4C, HPA028927-Sigma). Cells were washed 3 times for 10 min each in PBS followed by a 30–60 min incubation at room temperature with the secondary antibody diluted in blocking solution in the presence of DAPI (1:1000, 1 µg/ml, Sigma). As secondary antibody fluorescently labeled donkey anti rabbit-IgG (1:400, AlexaFluor ® 568 Thermo Fisher) was used. The three final washes for 5 min each in PBS followed by washing briefly first in $ddH_2O$ then absolute EtOH. Finally, coverslips were mounted using ProLong® Gold Antifade Mountant (Thermo Fisher). Images were acquired using the Zeiss LSM 780 Spinning Disk and were processed in Fiji[69].

### Apoptosis assay

For the detection of early apoptosis, an apoptosis kit with Annexin V-APC and propidium iodide (PI) (Biolegend) was used in combination with flow cytometry. Cells were seeded into coated 12-well plates at a density of $(0.5–1) × 10^6$ cells in 2 ml medium per well and infected either with the RCOR2 shRNA expressing lentiviral vectors or the scramble shRNA expressing control lentiviral vector as described in Competitive growth assay. Next day, cell culture media were replaced with fresh media. Two days after, cells were selected by 2 ug/ml Puromycin (InvivoGen) for two days. After puromycin selection cells were kept another day in the culture. Then, all floating and attached cells were harvested by Accutase and washed with PBS. Cells were then stained with Annexin V-APC and PI diluted in annexin-binding buffer according to the manufacturer's instructions. Samples were analyzed using a FACS BD Fortessa and FlowJo v10.8.2 software (BD Biosciences). After gating GFP + cells from single cell gate, quadrant gate was applied (for details see Source Data). Three independent experiments without technical replicates were performed and averaged for data presentation. Statistical analysis was performed using Prism 8 (GraphPad Prism).

### RCOR2 shRNA KD and differential gene expression analysis

EP1NS and EPD210FH cells were infected either with the RCOR2 shRNA expressing lentiviral vectors or the scramble shRNA expressing control lentiviral vectors as described in Apoptosis assay. Five days post infection, cells were harvested by Accutase treatment. Total RNA was isolated from these cells by the Maxwell® RSC simplyRNA Tissue Kit, used with the Maxwell® RSC Instruments (Promega). Gene expression profiles of transduced EP1NS and EPD210FH cells were generated on Affymetrix GeneChip human Genome U133 Plus 2.0 (U133v2) arrays. Differentially expressed genes analysis between RCOR2 KD and control samples was performed with limma R package[56].

### CTCF binding site inhibition in ST-ZFTA cells

EP1NS cells were expanded as described above and harvested by Accutase treatment (Sigma-Aldrich). Then, 5 µg/ml protamine sulfate (Sigma-Aldrich) and concentrated virus re-suspended in the respected medium together with $(1–2) × 10^6$ cells were seeded in to Geltrex coated 6-well cell culture plates and centrifuged at $700 × g$ for 45 min. and then incubated at 37 °C overnight. In the case of transduction with the pL.CRISPR.puro construct, cells were selected with 2 µg/ml puromycin (Invivogen) for 2–3 days, starting on the third day after transduction. The transduced cells were harvested 10 days after transduction and half of the cells were used for genomic DNA isolation (DNeasy Blood and Tissue Kit, Qiagene), the other half was used for RNA extraction (RNeasy Kit, Qiagene). First, gRNA efficiency was

estimated by PCR amplification of the site around the cut, followed by bulk PCR Sanger sequence analysis performed using the Synthego ICE analysis tool (https://ice.synthego.com/#/) according to their provided protocol. Secondly, cDNAs were synthesized by QuantiTect Reverse Transcriptase Kit (Qiagene) according to the manufacturers protocol using 500 ng total RNA input to check expression levels of ARL4C and NELFB. List of gRNAs and primers for amplification of the sites around the cuts can be found in Supplementary Data 6.

### Sample collection

Tumor samples were obtained with consent from patients with ependymoma under a protocol approved by the Institutional Review Board of UC San Diego (#171361). Study design and conduct complied with all relevant regulations regarding the use of human study participants. The study was conducted in accordance to the criteria set by the Declaration of Helsinki.

### Reporting summary

Further information on research design is available in the Nature Portfolio Reporting Summary linked to this article.

## Data availability

The data that support this study are available from the corresponding author upon reasonable request. The sequencing data raw materials (Hi-C, CTCF, WGBS) generated in this study have been deposited in the European Genome-phenome archive (https://www.ebi.ac.uk/ega/home) under the accession code: EGAS00001002696; this source already contains other data types (RNA-seq, H3K27ac) for the corresponding target tumor samples. The Affymetrix data used in this study are available in the GEO database under accession codes GSE64415, GSE50161, GSE50385, GSE21687, GSE3526. The EGA dataset is available under restricted access due to ethical controls and requires an approval for application in scientific research. The data access request should be sent directly by e-mail to daco-dkfz-b06x@dkfz-heidelberg.de and response will be provided in approximately two weeks. Source data are provided with this paper.

## Code availability

Scripts for processing the raw data and generating the figures as well as links to interactive genome view sessions are available via GitHub repository: https://github.com/kokonech/EPN_HiC_analysis.

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

## Acknowledgements

This work is supported by a generous endowment by the Clayes foundation to the Research Center for Neuro-Oncology and Genomics within the Rady Children's Institute for Genomic Medicine, a Hannah's Heroes St. Baldrick's Scholar Award (LC) and funding from the NIH National Institute of Neurological Disorders and Stroke Institute R21 NS116455 (LC), R21 NS120075 (LC) and R21 NS130137 (LC), the NIH National Cancer Institute U01 CA217885 (JPM), U01 CA184898 (JPM), U24 CA210004 (JPM and JTR); the NIH National Institute of General Medical Sciences R01GM074024 (JPM); and the NIH National Library of Medicine T15LM011271 (OC). The pediatric brain profiling at NYU in M.S. laboratory was in part supported by grants from the Friedberg Charitable Foundation and the Making Headway Foundation. Work in the laboratory of J.R.D. was supported by a NIH Early Independence Award (DP5 OD023071). This work used the Extreme Science and Engineering Discovery Environment (XSEDE), which is supported by National Science Foundation grant number ACI-1548562. We are thankful to David T. Jones for providing pediatric patient-derived SU-pcGBM2 cells. The INFORM program is financially supported by the German Cancer Research Center (DKFZ), the German Cancer Consortium (DKTK), the German Federal Ministry of Education and Research (BMBF), the German Federal Ministry of Health (BMG), the Ministry of Science, Research and the Arts of the State of Baden-Württemberg (MWK BW); the German Cancer Aid (DKH), the German Childhood Cancer Foundation (DKS), RTL television, the aid organization BILD hilft e.V. (Ein Herz für Kinder) and the generous private donation of the Scheu family.

## Author contributions

K.O., A.Ca., M.K., and L.C. prepared the manuscript and figures. K.O., O.C., A.Ch., M.P. J.Ro, E.F.J., and A.S. performed data analysis and visualization. A.Ca., S.W., D.E.P., A.J., and J.M.H. performed experimental validations, N.C., S.S., M.L., D.M., S.N., M.M., L.M., and M.S. processed and analyzed tumor material, R.B., S.C., K.K., R.A.H., K.S., and D.R. generated Hi-C libraries from frozen and FFPE tumor material and cell lines, K.W.P., T.M., N.J., P.F., P S.M., A.S., H.C., J.Cr., R.W.R., T.B.D., J.Co., G.M., A.S., K.A.M., S.Ki., C.H., S.Ku., M.D.T., J.Ri, J.M., S.M.P., F.A., S.M., F.B., G.F. and J.D. contributed to the study design and data interpretation. L.C. designed the study and L.C. and M.K. co-supervised the project.

## Competing interests

Derek Reid, Kristin Sikkink and Anthony Schmitt are employees of Arima Genomics, Inc. The remaining authors declare no competing interests.

## Additional information

Konstantin Okonechnikov[1,2,33], Aylin Camgöz[1,2,3,33], Owen Chapman[4], Sameena Wani[4], Donglim Esther Park[5,6], Jens-Martin Hübner[1,2], Abhijit Chakraborty[7], Meghana Pagadala[4], Rosalind Bump[8], Sahaana Chandran[8], Katerina Kraft[9], Rocio Acuna-Hidalgo[10,11], Derek Reid[12], Kristin Sikkink[12], Monika Mauermann[1,2], Edwin F. Juarez[4], Anne Jenseit[1,2,13], James T. Robinson[4], Kristian W. Pajtler[1,2,14], Till Milde[1,14,15], Natalie Jäger[1,2], Petra Fiesel[1,16], Ling Morgan[4], Sunita Sridhar[4], Nicole G. Coufal[6,17], Michael Levy[18], Denise Malicki[19], Charlotte Hobbs[20], Stephen Kingsmore[20], Shareef Nahas[20], Matija Snuderl[21,22], John Crawford[23], Robert J. Wechsler-Reya[6,17,24], Tom Belle Davidson[25], Jennifer Cotter[25], George Michaiel[25], Gudrun Fleischhack[26], Stefan Mundlos[10], Anthony Schmitt[12], Hannah Carter[4], Kulandaimanuvel Antony Michealraj[27], Sachin A. Kumar[27], Michael D. Taylor[27], Jeremy Rich[5,6,28], Frank Buchholz[3,29,30], Jill P. Mesirov[4,31], Stefan M. Pfister[1,2,14], Ferhat Ay[7,17], Jesse R. Dixon[8], Marcel Kool[1,2,32,34] & Lukas Chavez[4,20,24,31,34] ✉

[1]Hopp Children's Cancer Center (KiTZ), Heidelberg, Germany. [2]Division of Pediatric Neurooncology, German Cancer Research Center (DKFZ) and German Cancer Consortium (DKTK), Heidelberg, Germany. [3]National Center for Tumor Diseases (NCT): German Cancer Research Center (DKFZ) Heidelberg, Faculty of Medicine and University Hospital Carl Gustav Carus, Technische Universität Dresden, Helmholtz-Zentrum Dresden-Rossendorf (HZDR), Dresden, Germany. [4]Division of Genomics and Precision Medicine, Department of Medicine, University of California San Diego (UCSD), San Diego, USA. [5]Division of Regenerative Medicine, Department of Medicine, University of California, San Diego, La Jolla, CA 92037, USA. [6]Sanford Consortium for Regenerative Medicine, 2880 Torrey Pines Scenic Drive, La Jolla, CA 92037, USA. [7]Centers for Cancer Immunotherapy and Autoimmunity, La Jolla Institute for Immunology, La Jolla, CA, USA. [8]Peptide Biology Labs, Salk Institute for Biological Studies, La Jolla, CA, USA. [9]Center for Personal Dynamic Regulomes, Stanford University, Stanford, CA, USA. [10]Max Planck Institute for Molecular Genetics, Berlin, Germany. [11]Institute for Medical Genetics and Human Genetics, Charité Universitätsmedizin Berlin, Berlin, Germany. [12]Arima Genomics, Inc, San Diego, CA 92121, USA. [13]Faculty of Biosciences, Heidelberg University, Heidelberg, Germany. [14]Department of Pediatric Oncology, Hematology and Immunology, Heidelberg University Hospital, Heidelberg, Germany. [15]CCU Pediatric Oncology, German Cancer Research Center (DKFZ) and German Cancer Consortium (DKTK), Heidelberg, Germany. [16]CCU Neuropathology, German Cancer Research Center (DKFZ) and German Cancer Consortium (DKTK), Heidelberg, Germany. [17]Department of Pediatrics, University of California, San Diego, San Diego, CA 92093, USA. [18]Neurosurgery, University of California San Diego – Rady Children's Hospital, San Diego, CA 92123, USA. [19]Pathology, University of California San Diego – Rady Children's Hospital, San Diego, CA 92123, USA. [20]Rady Children's Institute for Genomic Medicine, San Diego, CA 92123, USA. [21]Department of Pathology, NYU Langone Health, NYU Grossman School of Medicine, 550 First Ave, New York, NY 10016, USA. [22]Laura and Isaac Perlmutter Cancer Center, NYU Langone Health, New York, NY, USA. [23]Department of Neurosciences, University of California San Diego – Rady Children's Hospital, San Diego, CA 92123, USA. [24]Tumor Initiation and Maintenance Program, NCI-Designated Cancer Center, Sanford Burnham Prebys Medical Discovery Institute, La Jolla, CA, USA. [25]Division of Hematology-Oncology, Cancer and Blood Disease Institute and Department of Pediatrics, Children's Hospital Los Angeles, Los Angeles, California, USA. [26]German Cancer Consortium (DKTK), West German Cancer Center, Pediatrics III, University Hospital Essen, Essen, Germany. [27]Division of Neurosurgery, Arthur and Sonia Labatt Brain Tumor Research Center, Hospital for Sick Children, University of Toronto, Toronto, ONT, Canada. [28]Department of Neurosciences, School of Medicine, University of California San Diego, La Jolla, CA 92037, USA. [29]Medical Systems Biology, Medical Faculty and University Hospital Carl Gustav Carus, TU Dresden, 01307 Dresden, Germany. [30]German Cancer Research Center (DKFZ), Heidelberg and German Cancer Consortium (DKTK) Partner Site Dresden, Dresden, Germany. [31]Moores Cancer Center, University of California San Diego (UCSD), La Jolla, CA, USA. [32]Princess Máxima Center for Pediatric Oncology, Utrecht, The Netherlands. [33]These authors contributed equally: Konstantin Okonechnikov, Aylin Camgöz. [34]These authors jointly supervised this work: Marcel Kool, Lukas Chavez. ✉e-mail: lukaschavez@health.ucsd.edu

