## [Peer Review File · Nature Communications]

3D genome mapping identifies subgroup-specific chromosome conformations and tumor-dependency genes in ependymomaEditorial Note: This manuscript has been previously reviewed at another journal that is not operating a transparent peer review scheme. This document only contains reviewer comments and rebuttal letters for versions considered at *Nature Communications*. Mentions of prior referee reports have been redacted. Mentions of the other journal have been redacted.

REVIEWERS' COMMENTS

Reviewer #1 (Remarks to the Author):

reviewer 1

There are interesting aspect of the revised manuscript. However, it still suffers from over-interpretation and claims of novelty (see below in bold and a point-by-point reply)

[redacted]

Reviewer 1 reply: I am not convinced about the usage of “novel” in structural variants (title) – and remain unconvinced about the (likely co-incidental) SVs near LAMC1. Moreover, without further functional analysis, these gene-regulatory dependencies remain putative

[redacted]

Reviewer 1 reply: The approach of requiring both Hi-C and WGS for SV calling is somewhat unintuitive. Hi-C has advantages over WGS and vice-versa. Line 143 suggests that Hi-C was used for SV calling, which was verified by WGS-based SV calling (line 150)

[redacted]

Reviewer 1 reply: Ok

[redacted]

Reviewer 1 reply: Ok

[redacted]

Reviewer 1 reply: What is the p-value provided in Fig 3c based on? Specific test should be performed comparing specific paired bins

[redacted]

Reviewer 1 reply: Ok

[redacted]

Reviewer 1 reply: I remain unconvinced that LAMC1 is not merely a dosage-gene. Fig S5h shows an expression increase from ~25 RPKM to ~37 RPKM of LAMC1 (~1.5-fold) in 1q-gain samples, which matches exactly a duplication, leading to an additional DNA copy of the

gene.

[redacted]

Reviewer 1 reply: I appreciate this analysis. Can the authors also please include the CTCF ChIP-seq data in Fig 5e.

The authors should perform the same analysis on ZFTA samples to demonstrate specificity. The expression and chromatin landscape appear similar for PFA and ZFTAs.

In this respect, I find it unlikely that there are no ZFTA loops at all in the genomic view. If this is the case, the authors should state this explicitly.

A new set of analysis has been performed on NELFB. Why did the authors not also delete the CTCF binding site in ZFTA – again, to show specificity. In addition to the chromatin landscape, the expression of NELFB is almost similar for PFA and ZFTA (Fig S6o)

reviewer 3

[redacted]

reviewer 3 reply: The fact that these are rare tumors does not merit circumstantial evidence and there is not sufficient evidence for any of these SVs in driving LAMC1 expression changes. LAMC1 is unconvincing

Reviewer #2 (Remarks to the Author):

The authors have sufficiently addressed my concerns. I still find the scope of this paper to be overly large. But I am supportive of publication at this point.

Point-by-point replies to review Nature Communications [redacted]

Reviewer 1

There are interesting aspect of the revised manuscript. However, it still suffers from over-interpretation and claims of novelty (see below in bold and a point-by-point reply).

[redacted]

Reviewer 1 reply: I am not convinced about the usage of “novel” in structural variants (title) – and remain unconvinced about the (likely co-incidental) SVs near LAMC1. Moreover, without further functional analysis, these gene-regulatory dependencies remain putative

Reply to the reviewer: Based on the reviewers’ comments, we have now changed the title as follows:

New title: 3D genome mapping identifies subgroup-specific chromosome conformations and tumor-dependency genes in ependymoma.

[redacted]

Reviewer 1 reply: Ok

[redacted]

Reviewer 1 reply: The approach of requiring both Hi-C and WGS for SV calling is somewhat unintuitive. Hi-C has advantages over WGS and vice-versa. Line 143 suggests that Hi-C was used for SV calling, which was verified by WGS-based SV calling (line 150)

Reply to the reviewer: We agree that HiC has advantages in SV calling and may be able to detect SVs that are invisible or difficult to call in standard WGS. In this study, we present all SVs called by HiC as Supplementary Tables, however, we decided to focus on the reconstruction of neo-TADs and integration of the gene regulatory environment near SV breakpoints for selected SVs that we were able to validate by WGS.

[redacted]

Reviewer 1 reply: Ok

[redacted]

Reviewer 1 reply: Ok

[redacted]

Reviewer 1 reply: What is the p-value provided in Fig 3c based on? Specific test should be performed comparing specific paired bins

Reply to the reviewer: The corresponding p-value is representing the difference between paired bins and was computed using the R package DiffLoop (PMID: 29028898). We now name this tool in the extended legend of Figure 3c.

[redacted]

Reviewer 1 reply: Ok

[redacted]

Reviewer 1 reply: I remain unconvinced that LAMC1 is not merely a dosage-gene. Fig S5h shows an expression increase from ~25 RPKM to ~37 RPKM of LAMC1 (~1.5-fold) in 1q-gain samples, which matches exactly a duplication, leading to an additional DNA copy of the gene.

Reply to the reviewer: Based on the reviewer's comment, we have removed all LAMC1-related material from the main Figures. We have moved the corresponding Figures to the Supplementary Material and stress in the updated manuscript that a functional relationship between SVs and LAMC1 transcription was not demonstrated.

[redacted]

Reviewer 1 reply: I appreciate this analysis. Can the authors also please include the CTCF ChIP-seq data in Fig 5e.

Reply to the reviewer: We have now added the CTCF ChIP-seq data tracks into the updated Figure 4e (previously Figure 5, Figure 1 for reviewer #1 below). The CTCF ChIP-seq data at the functionally tested CTCF binding site is already available, together with the WGBS data, in Fig 4f.

Figure 1 for reviewer #1 (Figure 4e in the manuscript) Genome browser visualization of PFA ependymoma-specific DNA loops that associate two PFA enhancers (E1 and E2) with the ARL4C gene. Tracks for RNA-seq, H3K27ac, CTCF and Hi-C derived DNA loops are obtained by merging PFA (9EP1,9EP9, 7EP18) or ZFTA (11EP22, 4EP53, 7EP41), respectively.

The authors should perform the same analysis on ZFTA samples to demonstrate specificity.

Reply to the reviewer: We initially performed the same analysis in ZFTA tumors, but despite similar chromosomal landscapes, we did not identify genomic regions with hypermethylation and CTCF loss in ZFTA that fit our filtering criteria (as also described in the Methods section). We now state this explicitly in the manuscript Methods part accordingly.

New text: "The DMRs were overlapped with CTCF differential peaks (min adj. p-value 0.05) resulting in 1254 pairs. The analysis for gene selection was performed on hypermethylated CTCF loci for PFA (n=966) and ZFTA (n=43) using two different approaches. From both of them no results were found for ZFTA, but only for PFA."

The expression and chromatin landscape appear similar for PFA and ZFTAs. In this respect, I find it unlikely that there are no ZFTA loops at all in the genomic view. If this is the case, the authors should state this explicitly.

Reply to the reviewer: The Figure showed only DNA loops between the E1 enhancer and the ARL4C gene that span the CTCF binding site, which is present in ZFTA tumors but replaced by DNA methylation in PFA tumors. Since these long-range DNA loops are only present in PFA ependymoma tumors, the ZFTA HiC track was empty. Based on the reviewers' comment, we now include all DNA loops passing q-value 0.05 limit without additional filtering (Figure 1 for reviewer #1, Figure 4e in the manuscript).

A new set of analysis has been performed on NELFB. Why did the authors not also delete the CTCF binding site in ZFTA – again, to show specificity. In addition to the chromatin landscape, the expression of NELFB is almost similar for PFA and ZFTA (Fig S6o).

Reply to the reviewer: These experiments were actually already performed, confirming the effect for NELFB: the target CTCF loci was removed in ZFTA cell line (Suppl. Fig. 7l-p) and the expression of the gene strongly increased (Suppl. Fig. 7n). We have not performed the experiment in PFA ependymoma, as the CTCF binding site was replaced by hypermethylation in this subtype.

Indeed, the difference in mean values is rather small between PFA and ZFTA, but strong variance is also observed. This could be an impact of array limitations as well as variance among PFA subgroups. Nevertheless from additional inspection of NELFB expression in ependymoma RNA-seq data from INFORM data cohort (PMID: 34373263), more evident effect was observed as shown in the figure below.

Figure 2 for reviewer #1. Boxplot showing NELFB gene expression in PFA and ZFTA tumors (RNA-seq gene expression data for n=117 ependymoma tumors). The center line, box limits, whiskers and points indicate the median, upper/lower quartiles, 1.5× interquartile range and outliers, respectively. DEG limma p-val.: 1.107e-07.

reviewer 3

[redacted]

reviewer 3 reply: The fact that these are rare tumors does not merit circumstantial evidence and there is not sufficient evidence for any of these SVs in driving LAMC1 expression changes. LAMC1 is unconvincing

Reply to the reviewer: Based on the reviewers' comment, we have removed all LAMC1-related material from the main Figures. We have moved the corresponding Figures to the Supplementary Material and stress in the updated manuscript that a functional relationship between SVs and LAMC1 transcription was not demonstrated.

Reviewer #2 (Remarks to the Author):

The authors have sufficiently addressed my concerns. I still find the scope of this paper to be overly large. But I am supportive of publication at this point.